# Quality of Life after Desarda Technique for Inguinal Hernia Repair—A Comparative Retrospective Multicenter Study of 120 Patients

**DOI:** 10.3390/jcm12031001

**Published:** 2023-01-28

**Authors:** Mark Philipp, Matthias Leuchter, Ralph Lorenz, Eberhard Grambow, Clemens Schafmayer, Reiko Wiessner

**Affiliations:** 1Department of General, Visceral, Thoracic, Vascular and Transplantation Surgery, Rostock University Medical Center, 18057 Rostock, Germany; 2Institute for Implant Technology and Biomaterials-IIB E.V, Associated Institute of the University of Rostock, 18119 Warnemuende, Germany; 33+ Chirurgen, Berlin-Spandau, 13581 Berlin, Germany; 4Department of General and Visceral Surgery, Bodden-Kliniken Ribnitz-Damgarten, 18311 Ribnitz-Damgarten, Germany

**Keywords:** Desarda, TAPP, pure tissue, inguinal hernia, quality of life

## Abstract

Inguinal hernia repair, according to Desarda, is a pure tissue surgical technique using external oblique fascia to reinforce the posterior wall of the inguinal canal. This has provided an impetus for the rethinking of guideline adherence toward minimally invasive and mesh-based surgery of inguinal hernia. In this study, a retrospective analysis of this technique was conducted in two German hospitals. Between 6/2013 and 12/2020, 120 operations were performed. Analysis included patient characteristics, duration of operation, length of hospital stay, and perioperative complications. Data were used to achieve a matched-pair analysis comparing Desarda to laparoscopic transabdominal preperitoneal (TAPP) hernia repair. Propensity scores were calculated based on five preoperative variables, including sex, age, American Society of Anesthesiology classification, localization, and width of the inguinal hernia in order to achieve comparability. Additionally, we assessed pain level and quality of life (QoL) 12 months postoperatively. The focus of our study was a comparison of QoL to a reference population and TAPP cohort. The study population consisted of 106 male and 14 female patients, and the median age was 37.5 years. The median operation time was 50 min, and the median length of hospital stay was 2 days. At a follow-up of 17 months, the median recurrence rate was 0.8%, and two cases of chronic postoperative pain were recorded. Postoperative QoL does not significantly differ between Desarda and TAPP. In contrast, Desarda patients had a significantly higher QoL compared with the reference population. In summary, Desarda’s procedure is a good option as a pure tissue method for inguinal hernia repair.

## 1. Introduction

The repair of inguinal hernia has been modified in numerous ways over the last 100 years. Recent worldwide guidelines base their recommendations on meta-analyses and randomized control trials (RCTs), but they are still controversial [1]. The minimally invasive approach and mesh-based repair receive a strong recommendation, and it has become challenging to perform a mesh-free (pure tissue) technique in primary inguinal hernia regardless of sex, age, or other factors [2,3].

Despite guideline recommendations, the pure tissue repair of the inguinal canal has never lost popularity, especially in low-resource countries where mesh implants are rare and expensive, as availability is limited [4]. Nevertheless, even in high-income countries, there are still arguments for a renaissance of pure tissue repair, especially when addressing the problem of chronic pain and the disadvantages of foreign body implants [5,6]. A tailored approach to inguinal hernia repair should include a mesh-free option.

A novel approach for mesh-free inguinal hernia repair was introduced by M.P. Desarda in 2001 [7]. The term pure tissue evolved and has led to an intense debate among hernia surgeons [5]. The first self-reported results of M.P. Desarda were promising but based on a single surgeon’s experience [8]. In the course of defining a true pure tissue method, Desarda used only long-term resorbable sutures [9]. Since then, his intuitive technique has been characterized by the use of only autologous external oblique fascia and long-term absorbable sutures to stabilize the posterior inguinal wall in order to avoid chronic pain. This seems important, as numerous variations in inguinal hernia repair have been historically described and scientifically established, and even reinforcement strategies have used biological mesh material as pure tissue [10,11].

Although most well-trained hernia surgeons should be familiar with all techniques, we nowadays face the problem of the inferior ability to perform pure tissue procedures. [12]. Current guidelines propose Shouldice to be the preferred technique and state that Desarda needs further supporting data [1]. However, in the available studies, systematic reviews, and meta-analyses, Desarda is usually compared with Lichtenstein [13,14].

This study followed the implementation of Desarda’s operation in a tertiary institution, as it was previously published by our group [15]. It was designed as a retrospective study that aimed to demonstrate equivalent results compared with a proven and established technique and was conducted at two German hospitals. The study evaluated perioperative parameters next to patient-reported outcomes measures (PROMs) in the context of quality of life (QoL).

## 2. Materials and Methods

Based on the description of M.P. Desarda and available publications, we introduced Desarda’s repair in a tertiary hospital (Department of General, Visceral, Thoracic, Vascular and Transplantation Surgery, Rostock University Medical Center) in 2013 and extended it to a regional hospital in 2016 (Department of General and Visceral Surgery, Bodden-Kliniken Ribnitz-Damgarten). Anonymous data acquisition was based on patients’ written informed consent and permission for registry participation. The retrospective cohort study was approved by the Institutional Review Board (A2022-0128).

On behalf of our own experience, we initially offered the treatment to selected patients, including males under 40 years and females under 50 years, as well as to patients who had risk factors for chronic pain or prejudices toward implants. After successful implementation of the method, we expanded the inclusion criteria.

A total of 120 patients underwent inguinal hernia repair according to Desarda. All cases were performed as elective surgery. Since TAPP is the standard procedure for inguinal hernia repair in our institutions, a mesh-based operated cohort was used as a benchmark.

### 2.1. Operation Technique

The technique was adapted from Desarda’s description using long-term resorbable sutures only [9]. Following a conventional approach to the inguinal canal, an indirect hernia sac was ligated using Vicryl 2/0, preferably hiding the stump under the internal oblique muscle. A direct defect was minimized by gathering the transversal fascia using Vicryl 2/0. A relevant femoral hernia was precluded by digital exploration inferiorly of the inguinal ligament toward the vascular lacuna. The external oblique fascia was sewn using continuous PDS 2/0 to the basis of the inguinal ligament, starting at the pecten ossis pubis and continuing to the internal inguinal ring. A 2 cm wide strip of external oblique fascia was incised and left attached medially and laterally, thus forming a new posterior wall of the inguinal canal. The superior edge of the strip was fixed to the internal oblique muscle fibers using PDS 3/0, meticulously avoiding the hypogastric nerve.

### 2.2. Outcome Parameters

Parameters proving our assumption were the evaluation of the duration of the operative procedure, the learning curve, and the length of stay in hospital. Supplementary data reinforcing our study objectives were obtained by the associated hospital where this technique was implemented and later applied simultaneously.

The primary endpoint of the study was the rate of recurrence and PROMs, including pain level and quality of life (QoL), in the long-term follow-up. For QoL analysis, the validated German version of the EQ-5D Health Questionnaire was used. The EQ-5D descriptive system consisted of 5 dimensions (mobility, self-care, daily activities, pain or discomfort, and psychological state) in 5 levels (no problems, slight, moderate, severe, and extreme problems). We compared these items with their levels in a reference population that originates from an evaluation of a sample of 3552 persons [16] Numeric analog scales (NASs) were used to assess postoperative pain.

The follow-up was scheduled 12 months postoperatively and conducted either through compulsory postoperative appointment or postal questionnaire and included a survey regarding QoL. The survey was used and established in our institution with numerous other hernia patients [17]. Our reference cohort (TAPP) received follow-up questionnaire, including QoL survey, under identical conditions.

All postoperative complications were based on clinical symptoms and physical exam findings observed during the follow-up. Only if patients stated problems in their returned postal questionnaire were they invited for clinical examination.

### 2.3. Statistics

For continuous data with a normal distribution, means are presented with standard deviations. When data were not normally distributed, data are given as median with interquartile range (IQRs, i.e., 25th and 75th percentiles). The Kolmogorov–Smirnov test was performed to assess normality of data. The *p*-values for continuous outcome measurements with a normal distribution were calculated using the Kruskal–Wallis rank sum test. If data were not normally distributed, we used Mood’s median test to analyze significant differences between study groups. The Fisher exact test or the χ^2^ test was used to determine the significance of intergroup differences for categorical variables. Statistical reports and analyses were carried out using the statistic software “R!” [18]. Patients were matched using propensity scores incorporating multiple preoperative variables. To generate valid statistical comparison, we performed an exact matching. This technique matches each Desarda patient to all possible TAPP patients with exactly the same values on all the covariates. Hence, matched records will have identical characteristics except for their treatment status.

## 3. Results

### 3.1. Patient Characteristics

In the Desarda group, the median age was 37.5 years (range: 16–80) overall. For men, the median age was 38 years (range: 17–80), and for women, 32 years (range: 24–75). Compared with our standard group of inguinal hernia repair, the age was significantly different. In the TAPP group, the median age was 60 years (range: 19–87), clearly exemplifying our general patient recruitment. Median age was 60 years for males and 63 years for females (Figure 1).

Desarda’s procedure was performed on 120 patients, 106 males (86.5%) and 14 females (13.5%). A total of 246 patients were treated by TAPP (210 male (84%), 36 female (16%), *p* = 0.518). The ASA score in the Desarda group was I (*n* = 56), II (*n* = 52), and III (*n* = 12). The median BMI was 24.7 (range: 19.1–37.1) in the Desarda group. TAPP-treated patients had a median BMI of 25.7 (range: 17.9–42.7); for the *p*-values, see Table 1. In the Desarda group, we included 62 cases of right-sided and 58 cases of left-sided inguinal hernia. The type of hernia was direct/medially in 40 cases, indirect/laterally in 71 cases, and combined in 9 cases. Half of all patients had a defect size of <1.5 cm, 50 1.5–3 cm, and 10 >3 cm. Five patients in the Desarda group had had a prior operation and were classified as having a recurrent hernia.

### 3.2. Perioperative Outcome

The duration of the operative procedure was 50 min (median) for Desarda repair (range: 30–87), which was significantly shorter than for the transabdominal preperitoneal patch plasty (TAPP) procedure (median, 60 min; range: 21–160; *p* < 0.001) (Figure 2).

Desarda’s repair was performed throughout all stages of teaching and learning and still outperformed routine inguinal hernia surgery. Involving additional trainees increased the length of surgery, but after about 80 operations, the mean time decreased and stabilized at 50 min (see above). Eventually, the procedure was accomplished by seven surgeons; three surgeons reached the level of teaching the novel technique to fellow colleagues.

The median postoperative length of stay at hospital was shorter after hernioplasty by means of Desarda at 2 days (range: 0–8) compared to TAPP at 3 days (range: 0–15).

### 3.3. Quality of Life

The matched-pair analysis equalized potential distorting parameters. Included matching predictors were age, sex, ASA classification, location, and defect size (EHS classification) of the inguinal hernia (Table 2).

The time span of follow-up regarding QoL after the treatment was 17 months (IQR 12–22). The index of QoL postoperatively was significantly better in patients who underwent Desarda before matching. When aligning QoL data in propensity score matching, these significant differences vanish. Figure 3 shows the EQ-5D questionnaire scores after matched-pair analysis. The median in the treatment group was 0.999 and 0.959 in the TAPP group, leading to a *p*-value of 0.149. Significant differences between the Desarda patients compared to the standardized reference population are verifiable.

### 3.4. Follow-Up

In the immediate postoperative assessment, we found a significant (*p* < 0.001) higher pain level (NAS) on the first postoperative day in the Desarda group (4, IQR 2–5) compared with TAPP (3, IQR 2–3).

At a median follow-up of 17 months (range: 5–36), we found one recurrence (0.8%) and two patients with chronic (prolonged, >3 months) pain (1.7%) in the Desarda group. In one case, persisting pain could not finally be differentiated from preexisting hip arthrosis.

The extended questionnaire showed faster recovery after TAPP during the first 14 days after the operation. In addition, 50% of recovery (self-reported inability to work) was reached after 21 days following TAPP and 28 days following Desarda. After 60 days, 2% of the patients treated with the technique according to Desarda and 10% of the patients with a TAPP hernia repair were still unable to perform their work (Figure 4).

## 4. Discussion

Reasons for renewed interest in pure tissue repair for inguinal hernia are numerous. There are persisting concerns associated with implanted hernia meshes regarding chronic postherniorrhaphy pain, visceral complications following minimally invasive and mesh-based techniques, as well as long-term uncertainties toward later surgical procedures, e.g., radical prostatectomy [19,20,21].

Nevertheless, the Hernia Surge Guideline states a weak recommendation for pure tissue inguinal hernia repair [1]. One problem could be the consistency and standardization of the surgical technique, which remains the main risk factor for the failure of mesh-free inguinal hernia repair [12]. Prospectively, the individual advantages and risks remain, and the idea of a tailored approach to hernia surgery might be exemplified by the discussion about pure tissue repair regarding Desarda’s technique [22].

A systematic review including 14 randomized controlled trials (RCTs), though of a very heterogenous quality, overseeing 2791 patients concluded Desarda to be a valuable alternative to Shouldice with a need for further studies [23]. A few prospective and comparative studies show comprehensible results and include a comparison to Lichtenstein repair or the Bassini technique but lack long-term follow-up [24,25]. The first prospective data comparing Desarda to Lichtenstein included a 3-year follow-up and was published by Szopinski et al. [26]. In the context of the discussion toward a renaissance of pure tissue inguinal hernia repair, a few questions remain unclear, especially the selection of patients and a reliable long-time follow-up [27].

Despite the rather minimal evidence, we introduced the method of Desarda’s repair in our German university hospital in 2013 and initially reported preliminary results in 2015 [15]. The consecutive selection of suitable patients was originally intended to identify the ideal indication for a pure tissue technique in inguinal hernia repair, especially under the pressure and dominance of guideline-derived, mesh-based, and minimally invasive techniques in the Western world [10]. Our cohort study was designed to retrospectively follow up patients who underwent Desarda’s repair in a standardized setting.

The considerable difference in median age in our Desarda and TAPP cohort led to a relevant longer length of stay. It was adjusted by using propensity score matching. A further prospective trial should eliminate this bias.

By applying QoL questionnaires, we were able to demonstrate comparable outcomes regarding patient comfort, and in particular, the short-term advantages of minimally invasive mesh-based (TAPP) repair were leveled out when looking beyond a 180-day follow-up survey. The postoperative self-reported return-to-work analysis was comparable to the data from Szopinksi et al., with 28 days as the median. In the same Polish study group, the recurrence rate was 2%, and the rate of chronic pain was stated at 4.8% [26]. We identified fewer patients with chronic pain when using Desarda’s repair. From our knowledge, this study is the first to report on PROM and QoL after Desarda’s repair.

Selecting suitable patients to offer the Desarda procedure was at the discretion of the surgeon and, therefore, affected the outcome of our study. We were not able to identify the ideal patient, but we also did not see an exclusion criterion in age, sex, or BMI. We found an increasing interest in mesh-free techniques. Therefore, it is a limitation of our study that it was not randomized. We see these results as a basis for initiating a prospective randomized trial.

## 5. Conclusions

In an observational study to introduce the operation according to Desarda’s technique, we were able to show that the novel operation was successfully implemented. The results were equal, even in a low-volume prerequisite. This was underlined by an additional survey of the postoperative QoL, showing that Desarda was equal to TAPP and superior compared to the reference population.

## Figures and Tables

**Figure 1 jcm-12-01001-f001:**
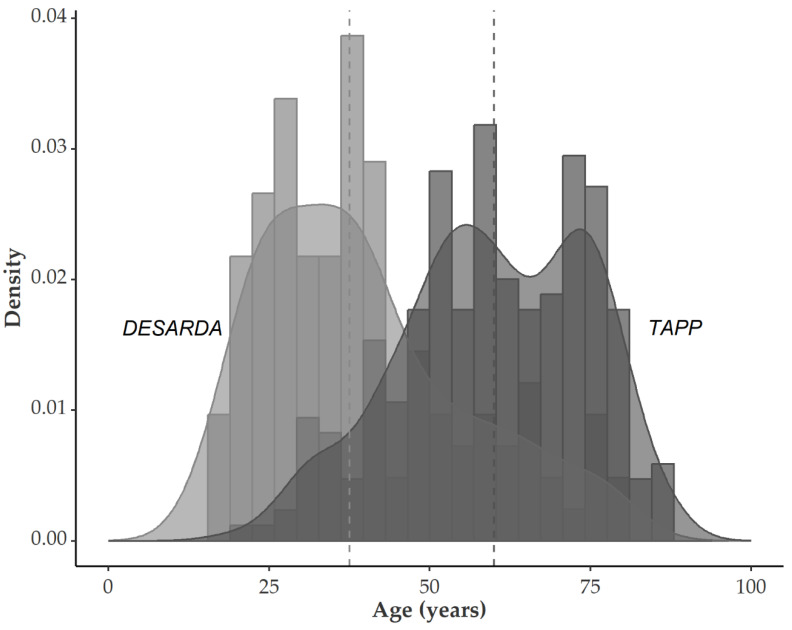
Distribution of age in patients undergoing inguinal hernia repair in our hospital, comparing transabdominal preperitoneal patch plasty (TAPP) to Desarda. The TAPP cohort reflects the age distribution of the patients in our tertiary hospital.

**Figure 2 jcm-12-01001-f002:**
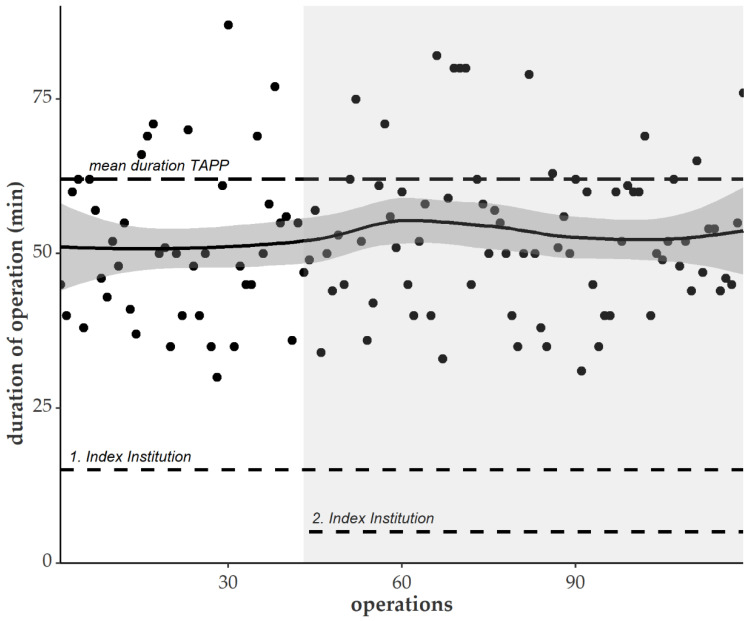
Correlation of operation time and number of procedures performed. The white area indicates single center results, and the area shaded light gray includes results from both hospitals. Dashed lines show mean duration of transabdominal preperitoneal patch plasty (TAPP) operation.

**Figure 3 jcm-12-01001-f003:**
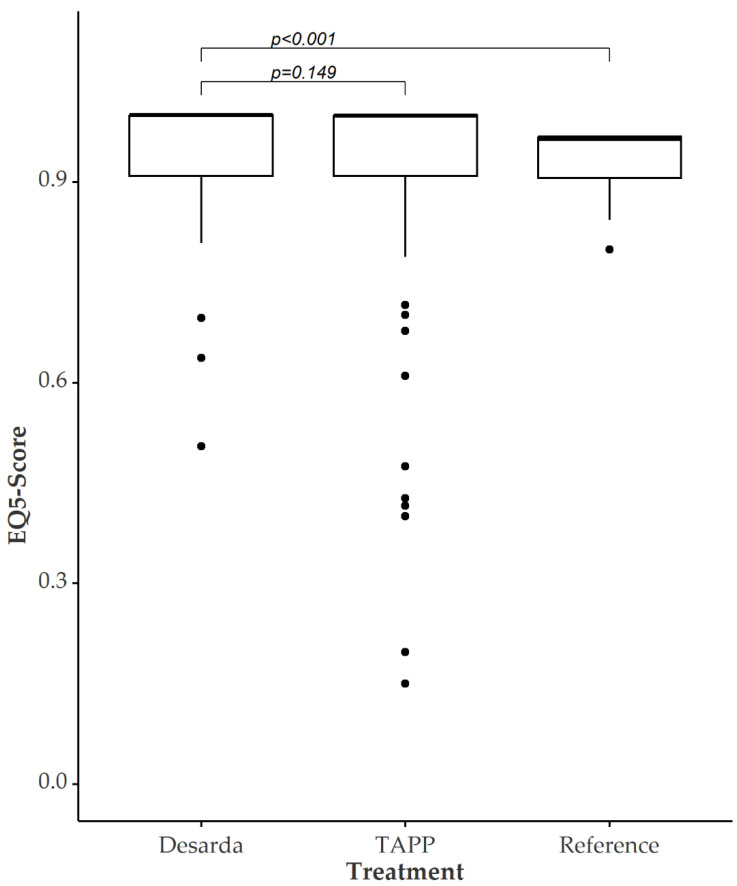
QoL after Desarda vs. TAPP vs. reference population after propensity score matching. *n* = 241 (Desarda = 98, TAPP = 143); data are given as median and IQR (25% and 75% percentile). Dots represent suspected outliers (≥1.5 IQR).

**Figure 4 jcm-12-01001-f004:**
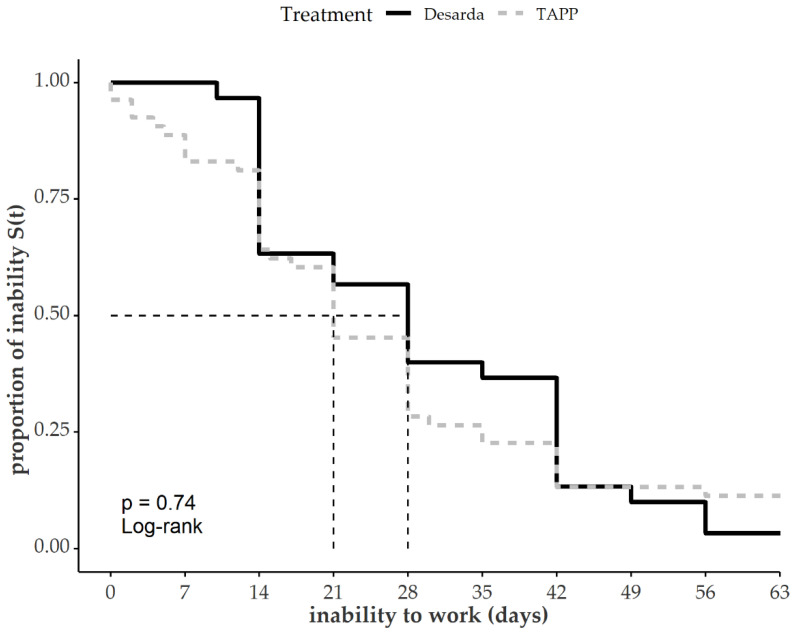
Self-reported time of inability to work after operation, *n* = 83.

**Table 1 jcm-12-01001-t001:** Baseline characteristics of unmatched patient cohorts by operative method. Data are given as *n* (%) or median (IQR). Chi-square test or Mood’s median test.

Characteristic	Desarda	TAPP	*p*
*n*	120	246	
*Age*	37.5 (26.8,49.0)	60 (51,73)	<0.001
**Sex**			0.518
*Male*	106 (88)	210 (85)	
*Female*	14 (12)	36 (15)	
*BMI*	24.6 (22.2,26.8)	25.7 (24.0,27.8)	0.126
*ASA score (I/II/III/IV)*	56/52/12/0	113/92/40/1	0.328
*Hernia side (right/left)*	62/58	127/119	1
**Type of hernia**			0.02
*Medial*	40 (33)	73 (30)	
*Lateral*	71 (59)	132 (54)	
*Combined*	9 (8)	41 (17)	
**Defect size**			0.007
*<1.5 cm*	60 (50)	85 (35)	
*1.5–3 cm*	50 (42)	145 (59)	
*>3 cm*	10 (8)	16 (7)	

TAPP: laparoscopic transabdominal preperitoneal; BMI: body mass index; ASA: American Society of Anesthesiologists Physical Status Classification System.

**Table 2 jcm-12-01001-t002:** Baseline characteristics of matched patient cohorts by operative method. Data are given as *n* (%) or median (IQR). Chi-square test or Mood’s median test.

Characteristic	Desarda	TAPP	*p*
*n*	98	143	
*Age*	43 (32.7,57)	43 (32.7,57)	1.000
**Sex**			1.000
*Male*	92 (94)	134 (94)	
*Female*	6 (6)	9 (6)	
*BMI*	25.1 (23.2, 26.8)	25.7 (23.8, 27,5)	0.359
*ASA score (I/II/III/IV)*	46/42/9	67/63/13	1.000
*Hernia side (right/left)*	51/47	73/70	0.896
**Type of hernia**			0.260
*Medial*	22 (22)	44 (31)	
*Lateral*	67 (68)	83 (58)	
*Combined*	9 (9)	16 (11)	
**Defect size**			0.993
*<1.5 cm*	52 (53)	76 (53)	
*1.5–3 cm*	43 (44)	63 (44)	
*>3 cm*	3 (3)	4 (3)	

TAPP: laparoscopic transabdominal preperitoneal; BMI: body mass index; ASA: American Society of Anesthesiologists Physical Status Classification System.

## Data Availability

The data presented in this study are available on request from the corresponding author. The data are not publicly available.

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
