# Peer review of "Quality of Life after Desarda Technique for Inguinal Hernia Repair—A Comparative Retrospective Multicenter Study of 120 Patients"

_jcm, 2023, doi:10.3390/jcm12031001_

Round 1

Reviewer 1 Report

Dear author, 

Great work, but there are points should be Clarified:

1-your title is not descriptive as it descripes Quality of life after Desarda technique and inside the manuscripte you are comparing  Desarda technique and TAPP.

2-you didn't mention if your cases were reccurent or not.

otherwise it is a good study and it deserves to be published 

goodluck 

Author Response

Dear Reviewer,

thank you very much for your kind reply and your useful explicit suggestions.

(2) We included the number of recurrent cases operated on. (line 155)

(1) We are aware of the retrospective nature of our study and did initially not intend to compare Desarda to TAPP. We used TAPP as a comparative cohort representing the standard procedure in our institution. We added the term comparative in the title to connect the title to the text. (line 3)

Reviewer 2 Report

This research article focused on the therapeutic evaluation of the Desarda technique in the treatment of inguinal hernia, standing out by the advantage of pure tissue surgery compared to currently popular clinical performance where artificial material is usually implanted.

In this paper the evidence is solid, the conclusion is the clear and the significance can promote clinical service. 

However, the median age in this research spanned around 20 years in this research. While this mismatch might result from the current retrospective cases, please still discuss the age difference, potential biases related to the other results, and future improvement in a prospective research in the Discussion Section.

Author Response

Dear Reviewer,

thank you very much for your kind reply and your reasonable remark.

The problem of the difference in age between both cohorts was very well in our mind. The retrospective design with a comparative cohort of TAPP should be improved by a further RCT. Currently we eliminated this issue using a propensity score matching and included this statement in the discussion chapter (line 253-255).